# PhyWakeNet: a dynamic wake model accounting for aerodynamic force oscillations

Xiaohao Liu<sup>a,b</sup>, Zhaobin Li<sup>a,b</sup>, and Xiaolei Yang<sup>a,b</sup>

<sup>a</sup>The State Key Laboratory of Nonlinear Mechanics, Institute of Mechanics, Chinese Academy of Sciences, Beijing, 100190, China

<sup>b</sup>School of Engineering Science, University of Chinese Academy of Sciences Beijing, 100049, China

Correspondence: Xiaolei Yang (xyang@imech.ac.cn)

Abstract. Advanced wind energy technologies require predictions of the dynamic behaviour of wind turbine wakes. In this work, we present a dynamic wind turbine model PhyWakeNet, a physics-integrated generative adversarial network-convolutional neural network (GAN-CNN) model for wind turbines under aerodynamic force oscillations. The model combines three interconnected submodels for the time-averaged wake, wake meandering, and small-scale wake turbulence. The time-averaged wake model derives from mass and momentum conservation based on the concept of momentum entrainment, which is computed based on the wake meandering and small-scale wake turbulence models. The wake meandering is captured through conditional GAN-reconstructed spatial modes and neural network-enhanced dynamic system for temporal evolution, while the small-scale wake turbulence is generated via a CNN based on the time-averaged wake, wake meandering, and inflow turbulence. Validation on wind turbine wakes under active control demonstrates the model's capability to predict frequency-dependent wake responses, velocity deficits, and turbulence kinetic energy. The model accurately captures temporal variations of key characteristics like instantaneous wake centers and velocity deficits, enabling potential applications in wake management to mitigate aerodynamic loads and power fluctuations in wind farms.

## 1 Introduction

Wind turbine wakes significantly impact wind farm performance by reducing power output, increasing aerodynamic loads, and contributing to power output fluctuations (Barthelmie and Jensen, 2010; Stevens and Meneveau, 2017; Meyers et al., 2022). Emerging advancements in wind energy technology Howland et al. (2022); Meyers et al. (2022) aim at active control of wind turbine wakes to mitigate their negative impacts. This presents new challenges to computational wake modeling, that not only the time-averaged statistics but also the dynamic behaviour of wind turbine wakes need to be captured. However, the modeling capabilities of existing wake models remain limited, with most of them developed for time-averaged wakes. One critical challenge is the incorporation of aerodynamic force oscillations, a critical factor triggering wake meandering Li et al. (2022a); Messmer et al. (2024a), the most important coherent flow structures in far wake. In this work, we propose a novel modeling framework that integrates physical principles with advanced machine learning techniques to predict the dynamic behaviour of wind turbine wakes under aerodynamic force oscillations.

40

50

Wind turbine wake modeling approaches range from computationally intensive large-eddy simulation (LES) to fast analytical models. LES directly resolves the energy-containing eddies in atmospheric turbulence while modeling subgrid-scale effects on the resolved flow field. For wind turbine wake simulations, blade aerodynamics is typically parameterized through forcing terms to mitigate computational loads (Li et al., 2022c). Despite these parameterizations, LES of wind turbine wakes still requires substantial computational resources, with simulation times extending from days to weeks depending on the required spatiotemporal resolutions and the spatiotemporal span of interest. This substantial computational demand renders LES currently impractical for wind energy project design and control optimization applications. Analytical wake models, typically derived from volume-integrated mass and momentum conservation equations, remain the most widely used approach in wind energy applications. The foundational model developed by Jensen Jensen (1983) represents the most prominent example, characterizing downwind velocity deficits through wake expansion modeling and approximating radial velocity deficit distributions using either Gaussian or cosine functions (Bastankhah and Porté-Agel, 2014; Xie and Archer, 2015; Tian et al., 2015). Intermediate-fidelity models have also been developed, exemplified by approaches solving simplified Navier-Stokes equations (Ainslie, 1988) and the vortex-based methods (Segalini and Alfredsson, 2013). These mid-fidelity models offer enhanced physical representation by directly resolving additional spatial dimensions, thereby eliminating the need for predefined wake shape assumptions. Despite their advantages, mid-fidelity models share a fundamental limitation with analytical wake models: neither approach can predict dynamic behaviour of wind turbine wakes.

The main coherent flow structure of interest for turbine-turbine interactions is wake meandering, a large-scale, low-frequency motion of wind turbine wake in the transverse directions. The most well-known wake meandering model is the dynamic wake meandering (DWM) model developed at Denmark University of Technology (DTU) (Larsen et al., 2008). The DWM model is based on the assumption that the wake can be treated as passive scalars advected by inflow large eddies with the employment of Taylor's frozen flow hypothesis (He et al., 2017). Scale-by-scale turbulence kinetic energy analysis showed that the inflow eddies with the integral length scale greater than  $\sim 3D$  (where D is the rotor diameter) are effective in advecting wind turbine wakes Zhang et al. (2023). The shear layer instability mechanism is another important mechanism for wake meandering. It has been systematically demonstrated using numerical simulations (Mao and Sørensen, 2018; Gupta and Wan, 2019; Li et al., 2022b), wind tunnel experiments (Messmer et al., 2024b; Schliffke et al., 2024) and field tests (Angelou et al., 2023). Blade aerodynamics, especially its temporal force oscillations, is a critical factor for the onset and the strength of wake meandering, and is becoming a novel principle for active wake control strategies Li et al. (2022b); Messmer et al. (2024b).

In this work, we develop a novel dynamic wake model that synergistically combines physical principles with machine learning methods to compute the spatiotemporal characteristics of wind turbine wakes under aerodynamic force oscillations. The proposed model integrates three interconnected submodels: 1) A time-averaged wake model; 2) A wake meandering model; 3) A model for small-scale turbulence. The proposed model is trained using comprehensive LES datasets and successfully applied to compute the dynamics of wind turbine wakes under transverse aerodynamic force oscillations.

#### 2 METHODS

The proposed PhyWakeNet model is based on the decomposition of the instantaneous velocity  $\mathbf{u}(\mathbf{x},t)$  as follows:

$$\mathbf{u}(\mathbf{x},t) = \overline{\mathbf{u}}(\mathbf{x}) + \widetilde{\mathbf{u}}(\mathbf{x},t) + \mathbf{u}''(\mathbf{x},t) \tag{1}$$

where  $\overline{\mathbf{u}}$ ,  $\widetilde{\mathbf{u}}$ , and  $\mathbf{u}''$  denote the time-averaged, wake meandering, and small-scale fluctuating velocity components, respectively. The model requires two primary inputs: the atmospheric flow conditions ( $C_{af}$ ), and the turbine operating conditions (particularly control actions for active wake control, also denoted as  $C_{op}$ ). The velocity field  $\mathbf{u} = \overline{\mathbf{u}} + \widetilde{\mathbf{u}} + \mathbf{u}''$  constitutes the model output. The time-averaged velocity field  $\overline{\mathbf{u}}(\mathbf{x})$  is derived from mass and momentum conservation principles. The wake meandering component  $\widetilde{\mathbf{u}}(\mathbf{x},t)$  is modeled through: 1) A conditional generative adversarial network (CGAN) for the dominant spatial modes, and 2) A data-driven dynamical system for temporal evolution. The small-scale velocity fluctuations  $\mathbf{u}''(\mathbf{x},t)$  are generated by a convolutional neural network (CNN) that takes both the inflow conditions, and time-averaged and wake meandering flow field as inputs. The coupling of the three submodels is enabled by both physical insights and machine learning methods. A key challenge is to quantify the enhanced wake-ambient flow mixing induced by active wake control strategies, which is modeled based on the momentum entrainment concept, quantifying the combined effects of wake meandering and small-scale velocity fluctuations on wake recovery. In the following of the paper, u, v, and v represent the streamwise, spanwise, and vertical velocity components, respectively. The fluctuating components are collectively denoted as  $\mathbf{u}'(\mathbf{x},t) = \widetilde{\mathbf{u}}(\mathbf{x},t) + \mathbf{u}''(\mathbf{x},t)$ .

## 2.1 Time-averaged wake model

## 2.1.1 Governing equations

The time-averaged wake flow model is formulated based on mass and momentum conservations, predicting both velocity deficit and wake width evolution along the wind turbine downstream direction. This model incorporates enhanced mass and momentum fluxes resulting from wake meandering and small-scale velocity fluctuations through an entrainment model. Specifically, the following mass and momentum conservation equations are employed,

$$\begin{cases}
\frac{\mathrm{d}(A_w \overline{u}_w)}{\mathrm{d}x} = V_e S_w \\
\frac{\mathrm{d}(A_w \overline{u}_w^2)}{\mathrm{d}x} = V_e S_w \overline{u}_a
\end{cases} \tag{2}$$

where  $A_w$  is the wake cross-sectional area normal to the centerline,  $\overline{u}_w$  is the mean streamwise wake velocity,  $S_w$  represents the wake-ambient flow interface area per unit downwind distance,  $V_e$  is the entrainment velocity,  $\overline{u}_a$  is the ambient mean streamwise velocity. The entrainment velocity  $V_e$  is computed through the entrainment coefficient E,

$$V_e = E(\overline{u}_a - \overline{u}_w),\tag{3}$$

where E quantifies the rate at which ambient fluid is entrained into the wake. The entrainment approach represents a well-established method for modeling the development of highly turbulent regions into relatively quiescent ambient flows (Morton

**Figure 1.** Schematic of the proposed PhyWakeNet model including three submodels for the time-averaged, meandering, and small-scale turbulence of wind turbine wakes. The inputs include the atmospheric flow conditions and the turbine operational conditions. The output is the spatiotemporal variation of velocity field. The time-averaged wake flow is modelled based on the mass and momentum conservations. The wake meandering and small-scale turbulence are modelled using GCAN and CNN. The impacts of wake meandering and small-scale turbulence on time-averaged wake are modelled based on the momentum entrainment model. The outputs from the time-averaged wake model and the wake meandering model are employed for the construction of small-scale turbulence.

et al., 1956). For wind turbine wake modeling specifically, it has been employed in the work by Luzzatto-Fegiz Luzzatto-Fegiz (2018). The wake's cross-sectional shape is modeled as an ellipse with major axis  $D_{w1}$  and minor axis  $D_{w2}$  to capture the directional effects of aerodynamic force oscillations on wake meandering preferences. Consistent with this elliptical assumption, we postulate that the wake growth rates along both principal directions scale with the ratio of their respective entrainment coefficients, while the entrainment coefficient itself follows an elliptical distribution. These considerations yield the following final governing equations:

$$\begin{cases} \frac{\mathrm{d}\left(\frac{\pi}{4}D_{w1}D_{w2}\overline{u}_{w}\right)}{\mathrm{d}x} = \int_{0}^{2\pi}E(\theta)(U_{a} - U_{w})\cos(\alpha - \theta)r\,\mathrm{d}\theta, \\ \frac{\mathrm{d}\left(\frac{\pi}{4}D_{w1}D_{w2}\overline{u}_{w}^{2}\right)}{\mathrm{d}x} = \int_{0}^{2\pi}E(\theta)(U_{a} - U_{w})U_{a}\cos(\alpha - \theta)r\,\mathrm{d}\theta, \\ \frac{\mathrm{d}D_{w1}}{\mathrm{d}x} \bigg/\frac{\mathrm{d}D_{w2}}{\mathrm{d}x} = E_{1}/E_{2}, \end{cases}$$
(4)

**Figure 2.** Schematic of the time-averaged wake flow model. Left panel shows the wake profile in the hub-height x-y plane, while the right panel displays the wake cross-section in the y-z plane. Arrows indicate ambient flow entrainment. The wake cross-section is modeled as an ellipse (right panel), with aerodynamic force oscillations assumed to act in the y-direction.

Here  $E_1$  and  $E_2$  denote the entrainment constants along the major and minor axes respectively, with the angular dependence  $E(\theta) = \sqrt{E_1^2 \cos^2 \theta + E_2^2 \sin^2 \theta}$ . The angles  $\alpha$  and  $\theta$ , which define the orientation relationships, are illustrated in figure 2.

To solve the governing equations, initial conditions for both the streamwise velocity and wake diameter at the near-wake position are required. In this work, these are determined using one-dimensional momentum theory:

$$\begin{cases} \overline{u}_w = (1 - 2a)\overline{u}_{\text{in}}, \\ D_{w1} = D_{w2} = D, \end{cases}$$
 (5)

evaluated at the 1D downstream position. Here  $\overline{u}_{in}$  represents the incoming wind speed (which may differ from the ambient wind speed  $\overline{u}_a$  for turbines operating in an array), and a denotes the axial induction factor. The induction factor relates to the thrust coefficient  $C_T$  through the expression  $a = \frac{1 - \sqrt{1 - C_T}}{2}$ .

It should be noted that the governing equations presented above only provide the mean velocity deficit. To characterize the spatial distribution, we assume that the isocontours of  $\overline{u}_w$  follow an elliptical pattern, with the velocity deficit profile described by a cosine function along the major and minor axes:

$$\overline{u}_c \cos(\pi D_c/D)$$
. (6)

The parameters  $\overline{u}_c$  and  $D_c$  are determined by enforcing conservation of mass and momentum fluxes before and after the transformation:

$$\begin{cases} A_w(1-\overline{u}_w) = \int \overline{u}_c \cos(\pi y/D_c) dA_{cy}, \\ A_w(1-\overline{u}_w)^2 = \int (\overline{u}_c \cos(\pi y/D_c))^2 dA_{cy}. \end{cases}$$
 (7)

Substituting the specific parameters yields the concrete form of these equations:

$$\begin{cases}
\frac{\pi}{4} (1 - \overline{u}_w) D_{w1} D_{w2} = \int_0^{D_c/2} \pi \overline{u}_c \cos(\pi y/D_c) (1 + D_{w2}/D_{w1}) y \, dy, \\
\frac{\pi}{4} (1 - \overline{u}_w)^2 D_{w1} D_{w2} = \int_0^{D_c/2} \pi \left( \overline{u}_c \cos(\pi y/D_c) \right)^2 (1 + D_{w2}/D_{w1}) y \, dy.
\end{cases} \tag{8}$$

Solving these equations leads to analytical expressions for  $\overline{u}_c$  and  $D_c$ :

$$\begin{cases}
\overline{u}_c = \frac{8(1 - \overline{u}_w)}{\pi + 2}, \\
D_c = \sqrt{\frac{\pi^2(\pi + 2)}{32(\pi - 2)}}D_{w1}.
\end{cases} \tag{9}$$

A note is that the wake width in this new distribution differs from that under a uniform distribution. With  $\overline{u}_{in}$  and a specified, the governing equations for the time-averaged wake statistics ( $\overline{u}_w$ ,  $D_{w1}$ ,  $D_{w2}$ ) form a closed system when combined with the entrainment coefficient model.

#### 2.1.2 Wake entrainment model

120

The detailed theoretical derivation of the estimation method for parameter E is given in this section. Ambient turbulence and wake shear layer constitute the primary drivers of mass and momentum entrainment across the wake boundary. This physical understanding leads to the following formulation for the total entrainment coefficient:

$$E = E_{\rm a} + E_{\rm s} = E_{\rm a} + \frac{\langle V_e A_\eta \rangle}{\langle V_{e,o} A_{n,o} \rangle} E_{s,o},\tag{10}$$

where  $E_{\rm a}$  and  $E_{\rm s}$  represent contributions from ambient turbulence and wake shear layer effects respectively. The angle brackets  $\langle \cdot \rangle$  indicate time-averaged quantities. Subscript  $_{o}$  denotes reference values corresponding to conditions without active wake control, obtainable through either numerical simulations or experimental measurements. The ambient turbulence component  $E_{\rm a}$  is treated as a known input parameter. The model accounts for enhanced entrainment through proportionality to both the entrainment velocity  $V_{e}$  and the wake-ambient interface area  $A_{\eta}$ , the latter being directly computed from the modeled flow fields  $\widetilde{\mathbf{u}}$  and  $\mathbf{u}''$ .

The entrainment velocity  $V_e$  remains the only quantity requiring modeling in this formulation. To approximate  $V_e$ , we first establish the wake boundary as the iso-surface of streamwise velocity deficit  $\Delta u$ . The material derivative of  $\Delta u$  at an arbitrary point in the flow field is given by:

$$\frac{D\Delta u}{Dt} = \frac{\partial \Delta u}{\partial t} + \mathbf{u} \cdot \nabla \Delta u. \tag{11}$$

At the wake boundary, where the material derivative vanishes, this relationship simplifies to:

$$0 = \frac{\partial \Delta u}{\partial t} + \mathbf{u}_{\eta} \cdot \nabla \Delta u,\tag{12}$$

where  $\mathbf{u}_{\eta}$  represents the velocity of the wake boundary. The entrainment velocity is subsequently defined as the relative velocity component normal to this boundary:

$$V_e = (\mathbf{u} - \mathbf{u}_n) \cdot \mathbf{e}_n,\tag{13}$$

with  $e_n$  denoting the unit normal vector to the wake boundary. By subtracting Equation 12 from Equation 11, we derive the following expression for  $V_e$ :

$$V_e = -\left[\frac{1}{|\nabla \Delta u|} \frac{D\Delta u}{Dt}\right]_{\eta}.$$
 (14)

While this formulation theoretically enables direct computation of  $V_e$ , practical implementation presents challenges due to both computational complexity and the frequent unavailability of instantaneous velocity deficit field snapshots.

In what follows, we demonstrate that the entrainment velocity can be approximated using the time derivative of the wake center position in the transverse direction. The entrainment velocity is first expressed as,

$$V_e = -\left[\frac{1}{|\nabla(\Delta u)|} \frac{D(\Delta u)}{Dt}\right]_n \tag{15}$$

$$= -\left[\frac{1}{\sqrt{\left(\frac{\partial(\Delta u)}{\partial x}\right)^2 + \left(\frac{\partial(\Delta u)}{\partial y}\right)^2}} \left(\frac{\partial(\Delta u)}{\partial t} + u\frac{\partial(\Delta u)}{\partial x} + v\frac{\partial(\Delta u)}{\partial y}\right)\right]_{\eta}.$$
 (16)

For slender wakes, where both the transverse velocity component v and the streamwise gradient  $\partial \Delta u/\partial x$  remain small, this expression simplifies to

$$V_e \approx -\left[\left(\frac{\partial(\Delta u)}{\partial y}\right)^{-1} \frac{\partial(\Delta u)}{\partial t}\right]_{\eta}.$$
(17)

The transverse wake center position is defined as,

$$y_c(x,t) = \frac{\int_{\eta_l(t)}^{\eta_u(t)} \Delta u(x,y,t) y \, dy}{\int_{\eta_l(t)}^{\eta_u(t)} \Delta u(x,y,t) \, dy},$$
(18)

where  $\eta_l(t)$  and  $\eta_u(t)$  denote the transverse coordinates of the lower and upper wake boundaries, respectively. Introducing the cumulative velocity deficit function  $F(y,t) = \int_{-\infty}^{\eta} \Delta u(x,y,t) \, \mathrm{d}y$ , this expression transforms to,

$$y_c(x,t) = \frac{F(y,t)y|_{\eta_l(t)}^{\eta_u(t)} - \int_{\eta_l(t)}^{\eta_u(t)} F(y,t) \, \mathrm{d}y}{F(y,t)|_{\eta_l(t)}^{\eta_u(t)}}.$$
(19)

Recognizing that  $F(\eta_l,t)=0$  by definition, we obtain the simplified form,

$$y_c(x,t) = \eta_u(t) - \frac{\int_{\eta_l(t)}^{\eta_u(t)} F(y,t) \, dy}{F(\eta_u(t),t)}.$$
 (20)

The temporal evolution of the wake center position follows from differentiation,

$$\frac{\mathrm{d}y_c(x,t)}{\mathrm{d}t} = \frac{\mathrm{d}\eta_u(t)}{\mathrm{d}t} - \frac{\mathrm{d}}{\mathrm{d}t} \left( \frac{\int_{\eta_l(t)}^{\eta_u(t)} F(y,t) \mathrm{d}y}{F(\eta_u(t),t)} \right). \tag{21}$$

Under the assumption that velocity deficit integrals remain approximately stationary, this simplifies to,

$$\frac{\mathrm{d}y_c(x,t)}{\mathrm{d}t} \approx \frac{\mathrm{d}\eta_u(t)}{\mathrm{d}t}.$$
 (22)

At the upper wake boundary, where  $\Delta u(\eta_u(t),t) = C$  remains constant, differentiation yields,

$$0 = \frac{\partial(\Delta u)}{\partial \eta_u} \frac{\mathrm{d}\eta_u}{\mathrm{d}t} + \frac{\partial(\Delta u)}{\partial t}.$$
 (23)

Combining Equations (17), (22), and (23), and assuming  $\partial(\Delta u)/\partial\eta_u \approx \partial(\Delta u)/\partial y$ , we derive the entrainment velocity approximation,

$$160 \quad V_e \approx \frac{\mathrm{d}y_c}{\mathrm{d}t}.\tag{24}$$

This leads to the final expression for the entrainment coefficient,

$$E = E_a + \frac{\langle (\mathrm{d}y_c/\mathrm{d}t)A_\eta \rangle}{\langle (\mathrm{d}y_c/\mathrm{d}t)_o A_{\eta,o} \rangle} E_{s,o} \approx E_a + \frac{(\mathrm{d}y_c/\mathrm{d}t)_{\max} \langle A_\eta \rangle}{[(\mathrm{d}y_c/\mathrm{d}t)_f]_{\max} \langle A_{\eta,o} \rangle} E_{s,o}. \tag{25}$$

In the second formulation, the instantaneous  $dy_c/dt$  is replaced by its temporal maximum to avoid computing the product with  $A_{\eta}$ . The reference quantities  $(dy_c/dt)_o$ ,  $A_{\eta,o}$ , and  $E_{s,o}$  are derived from LES data: the first two are computed directly from simulations, while  $E_{s,o}$  is obtained through least-squares fitting of the velocity deficit to Equation (2). Notably, E varies spatially in oscillating turbine wakes due to the downstream evolution of  $A_{\eta}$ .

## 2.2 Wake meandering model

The coherent flow structures in the wake, represented by the leading SPOD modes, are modeled using a CGAN model, with their temporal evolution captured by a data-driven dynamical system. Specifically, the coherent velocity  $\tilde{\mathbf{u}}$  is expressed as:

170 
$$\widetilde{\mathbf{u}}(\mathbf{C}_{\mathrm{af}}, \mathbf{C}_{\mathrm{op}}, \mathbf{x}, t) \approx \sum_{i=1}^{N} a_i(\mathbf{C}_{\mathrm{af}}, \mathbf{C}_{\mathrm{op}}, t) \mathbf{\Phi}_i(\mathbf{C}_{\mathrm{af}}, \mathbf{C}_{\mathrm{op}}, \mathbf{x}),$$
 (26)

where  $\Phi_i$  represents the SPOD modes and  $a_i$  denotes the corresponding temporal coefficients, with N being the number of leading SPOD modes employed for coherent flow construction. Both  $\Phi_i$  and  $a_i$  depend on  $C_{af}$ , the atmospheric flow condition, and  $C_{op}$ , the wind turbine operational condition. A schematic of the coherent wake flow model is shown in figure 3.

## 2.2.1 Model for Spatial Modes

This section presents the modeling approach for the SPOD modes  $\Phi_i$ . The conditional generative adversarial network (CGAN) generates the  $i^{\text{th}}$  SPOD mode for specified conditions  $C_{\text{af}}$  and  $C_{\text{op}}$  according to the following expression:

$$\mathbf{\Phi}_{i}\left(\mathbf{C}_{\mathrm{af}},\mathbf{C}_{\mathrm{op}},\boldsymbol{x}\right) = \mathbf{\Phi}_{\mathrm{NN}}\left(\mathbf{C}_{\mathrm{af}},\mathbf{C}_{\mathrm{op}},\mathbf{\Phi}_{i}^{1}\left(\mathbf{C}_{\mathrm{af}}^{1},\mathbf{C}_{\mathrm{op}}^{1},\boldsymbol{x}\right),\mathbf{\Phi}_{i}^{2}\left(\mathbf{C}_{\mathrm{af}}^{2},\mathbf{C}_{\mathrm{op}}^{2},\boldsymbol{x}\right),\cdots\right),\tag{27}$$

where  $\Phi_{NN}$  denotes the neural network model trained on multiple realizations of the  $i^{th}$  SPOD mode,  $\Phi_i^j$  ( $j=1,2,\cdots$ ), under different conditions  $C_{af}^j$  and  $C_{op}^j$ . The model uses  $C_{af}$  and  $C_{op}$  as input features. This formulation implicitly assumes that the

**Figure 3.** Conceptual diagram of the coherent wake flow model. The upper portion illustrates the generation of spatial modes while the lower portion shows the model for temporal evolutions.

*i*<sup>th</sup> mode depends exclusively on corresponding modes from various conditions, without explicit consideration of interactions with other modes.

The CGAN model for generating spatial modes comprises two components (figure 4): a generator and a discriminator. The generator accepts the operating conditions  $C_{af}$  and  $C_{op}$  as inputs and produces predicted spatial modes  $\Phi_{NN}$ . The discriminator evaluates input pairs consisting of operating conditions ( $C_{af}$ ,  $C_{op}$ ) and corresponding spatial modes ( $\Phi_{NN}$ ), outputting a binary classification (real or fake). During training, the discriminator's weights remain fixed while only the generator's weights undergo updates. After training completion, the generator functions as a surrogate model for predicting spatial modes under arbitrary atmospheric and operational conditions.

## 2.2.2 Temporal Evolution Model

This section describes the model for the temporal coefficients  $a_i$  of the SPOD modes. The temporal evolution of coherent flow structures is modeled through a dynamic system representation for  $a_i(C_{af}, C_{op}, t)$  expressed as:

$$\frac{\mathrm{d}a_i}{\mathrm{d}t} = f_i,\tag{28}$$

where  $f_i$  represents the forcing term modeled using a deep neural network (DNN). The forcing term construction involves two sequential steps: first generating sample temporal coefficients for each SPOD mode under specified conditions  $C_{af}$  and  $C_{op}$ , followed by constructing the forcing term using these generated coefficients. The sample temporal coefficients derive from corresponding frequency spectra models for each SPOD mode, which are themselves modeled using neural networks trained

210

Figure 4. Schematic of the CGAN model for generating spatial modes.

on frequency spectra datasets across various operational conditions:

$$S_{a_i}\left(\mathbf{C}_{\mathrm{af}}, \mathbf{C}_{\mathrm{op}}, \omega\right) = \mathbf{DNN}_S\left(\mathbf{C}_{\mathrm{af}}, \mathbf{C}_{\mathrm{op}}, S_{a_i}^1\left(\mathbf{C}_{\mathrm{af}}^1, \mathbf{C}_{\mathrm{op}}^1, \omega\right), S_{a_i}^2\left(\mathbf{C}_{\mathrm{af}}^2, \mathbf{C}_{\mathrm{op}}^2, \omega\right), \cdots\right),\tag{29}$$

where  $\omega$  denotes frequency,  $S_{a_i}$  represents the frequency spectrum for the  $i^{th}$  SPOD mode under conditions  $C_{af}$  and  $C_{op}$ , and  $DNN_S$  constitutes the neural network model approximating the frequency spectrum. This model employs datasets of frequency spectra  $(S_{a_i}^1, S_{a_i}^2, \cdots)$  from various conditions while maintaining the same fundamental assumption as the SPOD mode model - that the frequency spectrum for specific conditions can be approximated using corresponding spectra from different conditions at the same modal order. The inverse Fourier transform of these learned frequency spectra yields the sample temporal coefficients for each SPOD mode.

Using the obtained sample temporal coefficients for leading SPOD modes, the forcing term is approximated through a deep neural network:

$$f_i = \text{DNN}_f(a_1, a_2, \cdots, a_N). \tag{30}$$

Notably, the forcing term for the  $i^{th}$  SPOD mode incorporates information from all considered SPOD modes' temporal derivatives ( $\dot{a}_i$ ) rather than relying solely on its own temporal derivative. This approach compensates for potential information loss at higher frequencies during neural network approximation of the frequency spectrum through DNN<sub>S</sub>. The resulting dynamic equation can be numerically integrated for arbitrary initial conditions, with this work employing the Runge-Kutta method described in Kennedy et al. (2000) for time integration.

215

230

240

## 2.3 Model for small-scale turbulence

To accurately approximate the entrainment constant for the time-averaged wake flow model, both coherent and incoherent turbulent fluctuations must be modeled. This section presents the incoherent wake flow model for generating incoherent turbulent fluctuations based on the time-averaged flows, coherent flows, and inflow conditions. The most straightforward approach is to incorporate higher-order modes directly during modal reconstruction. However, the complex spatial distribution and temporal variation of these higher-order modes make them difficult to predict, thereby compromising model predictability. To overcome this limitation, an alternative method has been developed based on physical insights and high-fidelity data.

A key physical insight suggests that within wind turbine wakes, small-scale structures tend to concentrate around the periphery of larger-scale wake structures. A schematic of the proposed incoherent wake flow model is shown in figure 5. By employing convolutional neural networks (CNNs) to predict these small-scale structures, we can simultaneously identify wake boundaries and augment small-scale structures. While a single snapshot of coherent structures can enrich small-scale representation, such predictions lack temporal evolution information, disrupting the connection between instantaneous small-scale states. To solve this issue, flow snapshots across time are employed to construct the model, resulting the following model for incoherent velocity fluctuations,

$$\mathbf{u}''(\mathbf{x},t) = \text{CNN}_{\mathbf{u}''}(\overline{\mathbf{u}}(\mathbf{x}) + \widetilde{\mathbf{u}}(\mathbf{x},t), \mathbf{u}_{\text{af}}(\mathbf{x},t_{\text{seq}})), \tag{31}$$

where  $\mathbf{u}_{af}(\mathbf{x},t_{seq})$  represents the velocity field of the ambient flow from the upstream measurement. The coordinate  $\mathbf{t}_{seq}$  denotes the snapshot sequence within the [-3D,0D] range rotor upstream. The predicted small-scale structures can be directly superimposed onto the large-scale flow field from the time-averaged wake flow model and coherent wake flow model at corresponding instants, yielding the complete instantaneous flow field.

Incorporating entire snapshot sequences (i.e., the  $\mathbf{u}_{af}(\mathbf{x},t_{seq})$  input for the  $\mathrm{CNN}_{\mathbf{u''}}$  model) during model training would significantly reduce efficiency and increase complexity. To address this, temporal downsampling is first applied to the snapshot sequences, substantially reducing memory requirements. The Taylor frozen hypothesis is then employed to reconstruct snapshots between sampling intervals, restoring temporal resolution while avoiding large-scale computational tasks.

# 235 3 RESULTS

## 3.1 Tests of submodels

This section evaluates various components of the proposed model. Momentum entrainment across the wake boundary serves as the key mechanism coupling the time-averaged wake flow model with the fluctuating wake flow model. Figure 6 presents the model-predicted wake–ambient interface area  $A_{\eta}$  and the entrainment velocity  $V_e$ , compared against LES results. Overall, good agreement is observed, especially the different streamwise evolutions under different force oscillating frequencies, although the model predictions are slightly lower. This discrepancy is considered acceptable, as the small-scale curled structures along the interface are challenging to capture accurately.

Figure 5. Schematic of the incoherent wake flow model.

Figure 6. Comparison of the wake-ambient interface area  $A_{\eta}$  and entrainment velocity  $V_e$  correspond to three different aerodynamic force disturbance characteristic frequencies of  $St_F=0.12$ ,  $St_F=0.25$ , and  $St_F=0.84$ .

This work is based on the fundamental assumption that the coherent flow component is predictable. To verify this assumption, we evaluate the model's performance in predicting leading SPOD modes for three characteristic aerodynamic force oscillation frequencies (St = 0.12, 0.25, and 0.84) in figure 7. As seen, our model demonstrates excellent performance across most cases,

Figure 7. Comparison of the first SPOD mode for the test cases with  $St_F = 0.12$  (a, d),  $St_F = 0.25$  (b, e), and  $St_F = 0.84$  (c, f), with (a-c) and (d-f) showing the results predicted by large-eddy simulation and the proposed model, respectively.

except for low-frequency conditions where coherent structures are less distinct. The model particularly excels at capturing the hub vortex formation, which produces a characteristic meandering pattern near the nacelle centerline in the highest frequency test case (St = 0.84). For the intermediate frequency case (St = 0.25), the simulation reveals a gradual downstream expansion of the meandering pattern. Conversely, the low-frequency case (St = 0.12) exhibits minimal spatial growth of the meandering pattern, a behavior that the model reproduces accurately. Overall, the results confirm the model's capability in predicting coherent wake dynamics under aerodynamic force oscillations in terms of: 1) Global flow pattern morphology; 2) Downstream evolution characteristics; and 3) Systematic variation with oscillation frequency.

The capability of the proposed model in predicting the energy spectra of SPOD modes is examined in Figure 8, comparing three configurations: (1) large-scale structures reconstructed from the first two modal orders without the incoherent wake

260

Figure 8. Comparison of the energy spectra of the leading SPOD mode correspond to three different aerodynamic force disturbance characteristic frequencies of  $St_F = 0.12$  (a),  $St_F = 0.25$  (b), and  $St_F = 0.84$  (c). In the figure, the black dashed line represents the  $k^{-5/3}$  law, while the gray solid line indicates the corresponding dimensionless frequency after temporal downsampling in the time domain. The red dashed lines and blue dash-dot lines represent the results with and without the inclusion of the incoherent wake flow model.

flow model, (2) large-scale structures combined with reconstructed small-scale turbulence using the incoherent wake flow model, and (3) reference LES results. The spectrum exhibits distinct peaks at St=0.25 and 0.84 in Figures 8(b) and (c), respectively, corresponding to the aerodynamic force oscillation frequency and dominant coherent flow structures. All three cases show an inertial subrange following the -5/3 power law. While the dominant peak frequency is well captured by the model without the incoherent wake flow model, the energy densities at other frequencies are significantly underpredicted and fail to exhibit the -5/3 scaling. With the inclusion of the incoherent wake flow model, the reconstructed flow field's energy spectra show excellent agreement with reference LES data across all frequencies in figure 8, extending even beyond the coarse sampling frequency (indicated by the gray line) used as input for the small-scale model. This demonstrates the model's remarkable generative capabilities. Furthermore, for the St=0.12 case, the energy density at the corresponding frequency is less pronounced compared to the other two cases. In contrast, the St=0.84 case reveals two harmonics of the fundamental frequency. The proposed model successfully captures these spectral variations with respect to aerodynamic force oscillation frequency.

At last, the performance of the wake flow model for small-scale fluctuations is tested. Figure 9 shows the comparison of the model-predicted small-scale velocity fluctuations with the LES results. Although the amplitudes of velocity fluctuations are somewhat underpredicted, two critical characteristics are well captured. They include: 1) the development of small scales, which initiate around the ambient-wake interface, grows in amplitude, and expands in the radial direction as traveling downstream; 2) the impacts of wake meandering on small-scale fluctuations, which follow the meandering pattern and are significantly amplified by the meandering motion;

Figure 9. Small-scale velocity fluctuations obtained from LES (a-e) and the proposed model (f-j) at the same instants. The contour is colored by instantaneous streamwise velocity. The three rows from top to bottom correspond to three aerodynamic force oscillation frequencies St=0.12, St=0.25, and St=0.84, respectively. The fourth and fifth rows correspond to two inflow conditions with  $[I_{\infty}=0.8\%, L_{\infty}=1.0D]$  and  $[I_{\infty}=0.2\%, L_{\infty}=4.0D]$ , respectively.

## 3.2 Time-averaged wake flow statistics

The section examines the time-averaged flow statistics predicted by the model. The quantitative evaluation of the proposed model's prediction of time-averaged wake statistics is presented in figure 11. We first examine the time-averaged velocity deficits  $\Delta \overline{u}$ . Although discrepancies exist in the shape of the velocity deficit in the near-wake region, the proposed model demonstrates strong predictive capabilities in the far-wake region, with predicted curves closely matching the reference profiles. The model accurately predicts differences in wake development for various aerodynamic force oscillations. Specifically, it captures the faster wind speed recovery observed for the two higher force oscillation frequencies (St=0.25 and 0.84). The overall agreement with reference profiles confirms the model's effectiveness in capturing the downwind wind speed recovery. This success stems from properly accounting for enhanced entrainment due to both coherent flow patterns and small-scale velocity fluctuations.

Figure 10. Comparison of the mean streamwise wake velocity  $\bar{u}_w$ , the major axis diameter  $D_{w1}$  and the minor axis diameter, correspond to three different aerodynamic force disturbance characteristic frequencies of  $St_F=0.12$ ,  $St_F=0.25$ , and  $St_F=0.84$ .

We first compare the model predictions of the mean streamwise velocity averaged over the wake's cross section and the minor and major axis diameters of the wake's cross section with the LES results. As seen in figure 10, the proposed model accurately captures the impacts of aerodynamic force oscillation frequencies on mean streamwise velocity and wake diameters. The wake recovers faster at the frequencies  $St_F = 0.25$ , 0.84 compared with  $St_F = 0.12$ . The streamwise velocity in the wake with  $St_F = 0.84$  is higher than the other two at 2D/3D turbine downstream locations. The wake flow with  $St_F = 0.25$ , on the other hand, starts its faster recovery at around 5D turbine downstream because of the onset of wake meandering.

We then examine the variance of the streamwise velocity fluctuations ( $\langle u'u'\rangle$ ) predicted by the proposed model. Overall good agreement with the reference data is observed, particularly for the case with St=0.25 where significant wake meandering occurs. The model demonstrates particular accuracy in predicting: 1) Locations of high-intensity  $\langle u'u'\rangle$  turbulent kinetic energy, which primarily occur near the blade tips; and 2) The overall magnitude of  $\langle u'u'\rangle$  fluctuations. For cases with St=0.12 and 0.84, where the wake lacks dominant coherent flow structures, the agreement with reference  $\langle u'u'\rangle$  data remains acceptable, though with larger discrepancies compared to the St=0.25 case. Overall, the model demonstrates strong capabilities in predicting basic wake flow statistics, including both the mean velocity deficit and streamwise velocity fluctuation variance. The following analysis focuses on evaluating the model's performance in predicting wake meandering statistics.

Figure 11. Time-averaged streamwise velocity deficit ( $\Delta \overline{u}$ , (a-c)) and variance of streamwise velocity fluctuations ( $\langle u'u' \rangle$ , (d-f)) profiles at various wind turbine downwind positions for three aerodynamic force oscillation frequencies (a, d) St=0.12, (b, e) St=0.25, and (c, f) St=0.84. Black solid lines: reference LES results; Dashed lines: model predictions for red  $\Delta \overline{u}$  and blue  $\langle u'u' \rangle$ .

Figure 12. Instantaneous flow fields obtained from LES (a-e) and the proposed model (f-j) at the same instants. The contour is colored by instantaneous streamwise velocity. The three rows from top to bottom correspond to three aerodynamic force oscillation frequencies St=0.12, St=0.25, and St=0.84, respectively. The fourth and fifth rows correspond to two inflow conditions with  $[I_{\infty}=0.8\%, L_{\infty}=1.0D]$  and  $[I_{\infty}=0.2\%, L_{\infty}=4.0D]$ , respectively.

## 3.3 Instantaneous wake flows

This section demonstrates the capability of the model in predicting instantaneous wake flows. We first compare the model-predicted instantaneous streamwise velocity fields against LES results in figure 12. The proposed model demonstrates strong agreement in capturing the onset of wake meandering, the large-scale meandering patterns across all tested locations, and the distinct wake behavior for different aerodynamic force oscillation frequencies. The model successfully reproduces small-scale flow structures that predominantly emerge along the wake boundary and surround the large-scale coherent structures. One limitation concerns the nacelle-induced flow fluctuations, that the near-wake centerline features are not captured. This is expected given the cosine-shaped velocity deficit assumption and the exclusion of nacelle effects and initial wake development physics in the model.

The amplitude of wake meandering  $\sigma_y$ , defined as the standard deviation of instantaneous wake center positions in the spanwise direction, is presented in Figure 13 for downstream locations x/D=5 and 10. The proposed model accurately predicts the variation of  $\sigma_y$  with respect to aerodynamic force oscillation frequency (St) and Atmospheric turbulence conditions At x/D=5,  $\sigma_y$  exhibits a maximum in the frequency range  $0.4 \le St \le 0.6$ , decreasing for both higher and lower frequencies. While the model captures this trend well, it shows slight overestimations of  $\sigma_y$  within this frequency range. Further downstream at x/D=10, the wake meandering amplitude  $\sigma_y$  displays a pronounced peak near St=0.3, with rapid decay at both higher and lower frequencies - a characteristic that the model reproduces with good fidelity. The analysis of inflow turbulence effects reveals that: 1) at x/D=5, the wake meandering amplitude is higher for higher inflow turbulence intensity; 2) at x/D=10, the sensitivity to inflow turbulence conditions diminishes significantly.

In active wake control applications, precise prediction of wake positions is essential. Figure 14 evaluates the model's performance in this regard by analyzing temporal variations of both spanwise wake center positions  $(y_c)$  and wake centerline velocity deficits  $(\Delta u_c)$  at the 10D downstream location. The proposed model demonstrates strong predictive capability, accurately capturing both long-term trends and short-term fluctuations in the wake behavior. While the agreement with reference data is generally good for both quantities, the predictions for  $y_c$  show better correspondence than those for  $\Delta u_c$ . This performance difference is partly caused the modeling framework's underlying assumptions: the time-averaged velocity deficit distribution for the wake is prescribed rather than simulated.

Figure 13. Comparison of actual and model-predicted wake center fluctuation amplitudes under varying aerodynamic force oscillation frequencies (a, b) and varying turbulent inflows (c, d). The subplots (a, c) show the comparison at a streamwise position of x/D = 5, while the subplots (b, d) show the comparison at a streamwise position of x/D = 10. The thirteen frequencies in (a, b) are 0.1, 0.12, 0.2, 0.25, 0.3, 0.4, 0.5, 0.6, 0.7, 0.8, 0.84, 0.9, and 1.0. The twelve inflows in (c, d) are the results of three turbulent integral length scales and four turbulent intensities. The crosses represent the unseen cases.

Figure 14. Comparison of temporal variations of spanwise wake center positions  $(y_c, (a-c))$  and wake centerline velocity deficits  $(\Delta u_c, (d-f))$  at the 10D downstream location. From top to bottom are three cases with motion frequencies of  $St_F = 0.12$ ,  $St_F = 0.25$ , and  $St_F = 0.84$ , respectively. The solid lines and the dashed lines represent the results of large-eddy simulation and the proposed model, respectively.

330

350

## 4 CONCLUSIONS

We proposed a physics-integrated GNN-CNN wake model (PhyWakeNet) for predicting the dynamics of wind turbine wakes under aerodynamic force oscillations. The PhyWakeNet model integrates three interconnected submodels: the time-averaged wake model, the wake meandering model, and the model for small-scale turbulence.

The time-averaged wake model is derived from the fundamental mass and momentum conservation principles, with its entrainment parameter dynamically determined based on the other two submodels. For wake meandering prediction, the model employs a spatiotemporal decomposition approach where the spatial modes are reconstructed through a combination of spectral proper orthogonal decomposition (SPOD) and conditional generative adversarial network (CGAN). Computational efficiency is maintained by retaining only the first five SPOD modes. Temporal evolution is captured through a dynamic system model enhanced by a deep neural network (DNN)-derived forcing term. The small-scale turbulence is generated by a convolutional neural network (CNN) that processes three key inputs: time-averaged wake field, wake meandering, and inflow turbulence. This comprehensive approach enables the model to capture a broad spectrum of wake dynamics.

Validation studies across various aerodynamic force oscillations and inflow turbulence conditions demonstrate the model's capabilities in capturing both the time-averaged and dynamic features of wind turbine wakes. The results show that the PhyWakeNet model accurately reproduces frequency-dependent variations in wake characteristics, outperforming existing engineering wake models in several aspects. Beyond predicting velocity deficits —a standard capability of traditional models —it successfully captures turbulence intensity distributions and the fluctuating wake features, including instantaneous wake positions and velocity deficits.

## 340 Appendix A: DATASETS

## A1 Numerical methods

The training datasets are generated using the large-eddy simulation module of the Virtual Flow Simulator (VFS-Wind) code Yang et al. (2015); Yang and Sotiropoulos (2018); Santoni et al. (2023). The flow physics is governed by the filtered incompressible Navier-Stokes equations:

$$\frac{\partial u_j}{\partial x_j} = 0$$

$$\frac{\partial u_i}{\partial t} + \frac{\partial u_i u_j}{\partial x_j} = -\frac{1}{\rho} \frac{\partial p}{\partial x_i} + \frac{\partial}{\partial x_j} \left( (\nu + \nu_t) \frac{\partial u_i}{\partial x_j} \right) + f_i$$
(A1)

where i, j = 1, 2, 3 denote spatial indices, u represents the velocity field, p is the pressure,  $\nu$  indicates the kinematic viscosity, and  $\nu_t$  stands for the eddy viscosity modeled through the Smagorinsky model with dynamically determined coefficients. The body force term  $f_i$  (per unit mass) originates from the actuator surface model, which captures both turbine blades and nacelle effects. Unlike the commonly used actuator line model, the actuator surface method explicitly incorporates blade geometry features, particularly the chord distribution along the spanwise direction, while also resolving nacelle geometry Yang and

Sotiropoulos (2018). Force and torque conservation during information transfer between the actuator surface grid and background flow solver grid is maintained through a smoothed discrete delta function approach Yang et al. (2009) using just 3 to 5 grid cells.

Spatial discretization employs a second-order central difference scheme, coupled with temporal advancement via a second-order fractional step method Ge and Sotiropoulos (2007). The momentum equation solution utilizes a matrix-free Newton-Krylov approach Knoll and Keyes (2004), while the pressure Poisson equation is solved through the Generalized Minimal Residual (GMRES) method accelerated by algebraic multi-grid techniques.

## A2 Simulated cases

In this study, we employ the NREL offshore 5 MW reference wind turbine model as our baseline configuration, which was developed by Jonkman, Butterfield, and Musial Jonkman et al. (2009). This turbine features a rotor diameter of 126 meters and a cuboidal nacelle measuring 2.3 meters by 2.3 meters by 14.2 meters.

Two distinct case configurations are investigated: one with inflow turbulence and one without. The tip-speed ratio  $\lambda$  is set at 7, while the Reynolds number based on inflow velocity and rotor diameter reaches approximately  $9.6 \times 10^7$ . The computational domain forms a cuboid measuring  $14D \times 7D \times 7D$  in the streamwise (x), horizontal (y), and vertical (z) directions respectively. The rotor is positioned 3.5D downstream from the inlet at the domain's central cross-section. A uniformly distributed inflow velocity is imposed at the inlet boundary (x=-3.5D), while the outlet boundary (x=10.5D) employs a Neumann condition  $\left(\frac{\partial u_i}{\partial x}=0\right)$ . For turbulent inflow cases, velocity fluctuations generated using the synthetic turbulence technique Mann (1998) are superimposed onto the uniform inflow profile. Lateral boundaries implement free-slip conditions throughout the simulations. The domain is discretized using a Cartesian grid with uniform spacing of  $\Delta x = D/20$  in the streamwise direction and  $\Delta y = \Delta z = D/20$  within the near-wake region  $(y,z \in [-1.5D,1.5D])$ . Grid spacing expands gradually outside this region. Comprising  $281 \times 141 \times 141$  nodes, this grid configuration has demonstrated capability for accurate predictions of velocity deficits and turbulence intensities in the turbine wake, as validated in our previous work Li et al. (2022b). Table A1 lists all simulated cases. Except for  $St_F = 0.12, 0.25, 0.84$ , all other cases are employed for model training.

## Appendix B: MODEL TRAINING DETAILS

## 375 B1 Training of the CGAN model for generating spatial coherent modes

The training process involves two competing components: the discriminator learns to distinguish between authentic pairs of spatial modes with their corresponding operating conditions, while the generator attempts to produce realistic spatial modes that create data pairs indistinguishable from genuine ones. The discriminator achieves this by minimizing its classification error. The objective function is expressed as:

$$\min_{G} \max_{D} V(D,G) = \mathbb{E}_{\mathbf{\Phi}_{n} \sim p_{\text{data}}(\mathbf{\Phi}_{n})}[\log D(\mathbf{\Phi}_{in}|C_{n})] + \mathbb{E}[\log(1 - D(G(\widehat{\mathbf{\Phi}}_{in}|C_{n}))]$$
(B1)

Table A1. Parameters for simulated cases

| Cases | Parameters                                                                                                                                                                 |  |  |  |  |  |
|-------|----------------------------------------------------------------------------------------------------------------------------------------------------------------------------|--|--|--|--|--|
| I     | Inflow turbulence: N/A                                                                                                                                                     |  |  |  |  |  |
|       | Force oscillation:                                                                                                                                                         |  |  |  |  |  |
|       | $\left(0.1, 0.12, 0.15, 0.2, 0.23, 0.25, 0.26, 0.3,\right)$                                                                                                                |  |  |  |  |  |
|       | $St_F \in \left\{ \begin{array}{l} 0.1, 0.12, 0.15, 0.2, 0.23, 0.25, 0.26, 0.3, \\ 0.4, 0.5, 0.6, 0.7, 0.8, 0.83, 0.84, 0.86, \\ 0.9, 1.0 \end{array} \right\}, A = 0.01D$ |  |  |  |  |  |
|       | 0.9,1.0                                                                                                                                                                    |  |  |  |  |  |
|       | Inflow turbulence:                                                                                                                                                         |  |  |  |  |  |
| II    | $I_{\infty} \in \{0.2, 0.4, 0.6, 0.8\}\%,$                                                                                                                                 |  |  |  |  |  |
| 11    | $I_{\infty} \in \{0.2, 0.4, 0.6, 0.8\}\%,$<br>$L_{\infty} \in \{1.0, 1.5, 4.0\}D$                                                                                          |  |  |  |  |  |
|       | Force oscillation: $St_F = 0.25$ , $A = 0.01D$                                                                                                                             |  |  |  |  |  |

In this formulation,  $\Phi_{in}$  represents an authentic sample drawn from the real data distribution  $p_{\text{data}}(\Phi_{in})$ ,  $C_n$  corresponds to the conditional vector, and  $D(\Phi_{in}|C_n)$  indicates the discriminator's estimated probability that  $\Phi_{in}$  constitutes a genuine sample under condition  $C_n$ . Since the distributions in the loss equation (B1) remain unknown, we employ empirical loss equations following Mirza and Osindero (2014). The hyperparameters for both generator and discriminator are detailed in Table B1.

Training data comprises flow snapshots from LES that capture spatial modes across various operational conditions. The conditional vector  $C_n$  originates from ambient flow and turbine operation parameters. Data preprocessing involves normalization and spatial mode alignment to maintain consistent input dimensions. The generated spatial modes form 3D tensors  $(191 \times 121 \times 5)$  representing five dominant spatial coordinates and flow variables.

## B2 Training of the DNN model for predicting the temporal evolution of coherent wake flows

**Frequency Spectrum Model.** The values of the hyperparameters are determined through validation errors using a systematic grid search approach. The employed hyperparameter values are presented in table B2.

Forcing term for the dynamic system. We generated 2000 snapshots from  $tU_{\infty}/D = 0$  to 10.8 through LES simulation. For different cases, we selected varying numbers of snapshots to maintain consistent periodicity across all datasets.

Our training data spans the interval from  $tU_{\infty}/D=0$  to 3.6, while data beyond  $tU_{\infty}/D=3.6$  serves as the test set, ensuring rigorous evaluation of the model's predictive capability on unseen data.

The DNN's performance critically depends on hyperparameter selection. We employed random search techniques to identify optimal hyperparameter configurations. The complete set of hyperparameters used is listed in table B4, while the optimal set obtained through random search appears in table B3.

In both tables B3 and B4,  $\sigma$  denotes the activation function,  $\alpha$  represents the learning rate, and  $\lambda$  is the regularization parameter. The variable  $n_{\text{iter}}$  indicates the number of iterations, while  $\beta_1$  and  $\beta_2$  correspond to the exponential decay rates in the Adam optimization method.

Table B1. Training details for the cGAN model

| Model                                             | Value/Description                                                      |  |  |  |  |
|---------------------------------------------------|------------------------------------------------------------------------|--|--|--|--|
|                                                   | Input: Noise vector $(z)$ and conditional feature $(c)$                |  |  |  |  |
|                                                   | Layers:                                                                |  |  |  |  |
|                                                   | • Linear $(z + \sin(2\pi c) + \cos(2\pi c) \rightarrow 1000)$          |  |  |  |  |
|                                                   | • BatchNorm1d (1000)                                                   |  |  |  |  |
| Generator                                         | • ReLU activation                                                      |  |  |  |  |
|                                                   | • Linear (1000 → 1000)                                                 |  |  |  |  |
|                                                   | • ReLU activation                                                      |  |  |  |  |
|                                                   | • Linear $(2000 \rightarrow 191 \times 5 \times 121)$                  |  |  |  |  |
|                                                   | Output: Generated image (191 $\times$ 5 $\times$ 121)                  |  |  |  |  |
|                                                   | Input: Image (191 $\times$ 5 $\times$ 121) and conditional feature (c) |  |  |  |  |
|                                                   | Layers:                                                                |  |  |  |  |
|                                                   | • Linear (img + $\sin(2\pi c) + \cos(2\pi c) \rightarrow 1000$ )       |  |  |  |  |
|                                                   | • ReLU activation                                                      |  |  |  |  |
| Discriminator                                     | • Linear (1000 → 100)                                                  |  |  |  |  |
|                                                   | • ReLU activation                                                      |  |  |  |  |
|                                                   | • Linear $(200 \rightarrow 1)$                                         |  |  |  |  |
|                                                   | Sigmoid activation                                                     |  |  |  |  |
|                                                   | Output: Probability of image being real (0 or 1)                       |  |  |  |  |
| Loss Function                                     | Binary Cross-Entropy Loss (BCELoss)                                    |  |  |  |  |
| Optimizer                                         | Adam                                                                   |  |  |  |  |
| Learning Rate (lr)                                | 0.0001                                                                 |  |  |  |  |
| Adam Parameters $\beta_1 = 0.9,  \beta_2 = 0.999$ |                                                                        |  |  |  |  |

Table B2. Hyperparameters for the Temporal Prediction Model

| DNN Architecture | σ    | α      | λ     | $oldsymbol{n}_{	ext{iter}}$ | $oldsymbol{eta_1}$ | $eta_2$ |
|------------------|------|--------|-------|-----------------------------|--------------------|---------|
| 1-500-1000-100   | Tanh | 0.0001 | 0.001 | 10,000                      | 0.9                | 0.999   |

**Table B3.** Model generation parameters for the Case St=0.25

| $n_{ m models}$ | $n_{ m layers_{min}}$ | $n_{ m layers_{max}}$ | $n_{ m hidden_{min}}$ | $oldsymbol{n}_{	ext{hidden}_{	ext{max}}}$ | $	heta_{ m min}$ | $oldsymbol{	heta}_{	ext{max}}$ |
|-----------------|-----------------------|-----------------------|-----------------------|-------------------------------------------|------------------|--------------------------------|
| 200             | 4                     | 12                    | 40                    | 240                                       | 8                | 12                             |

410

**Table B4.** Optimal hyperparameters for the case St=0.25

| DNN Architecture | σ   | α     | λ                        | $oldsymbol{n}_{	ext{iter}}$ | $oldsymbol{eta_1}$ | $oldsymbol{eta_2}$ |
|------------------|-----|-------|--------------------------|-----------------------------|--------------------|--------------------|
| 5-56-225-46-5    | ELU | 0.001 | $3.0128 \times 10^{-11}$ | 10,000                      | 0.9                | 0.999              |

## **B3** Training of the CNN model for predicting incoherent wake turbulence

We employ a three-dimensional convolutional neural network (3D-CNN) as our foundational architecture, as 3D-CNNs demonstrate exceptional capability in capturing complex patterns across both spatial and temporal dimensions. The model accepts a three-dimensional tensor input representing flow field data in space and time, and produces an output tensor of identical dimensions that predicts small-scale turbulence structures.

The training data originates from coarsely sampled turbulent flow fields. To implement the Taylor hypothesis, we define an advancing space-line that progresses with time. Behind this space-line, small-scale structures are obtained through interpolation of flow fields from subsequent time points within the coarse sampling interval. Ahead of the advancing line, small-scale structures derive from joint interpolation of flow fields from both preceding and subsequent time points within the sampling interval. Specific training parameters are detailed in Table B5.

**Table B5.** Training details for 3D-CNN model

| Parameter                | Value/Description                                                                                |
|--------------------------|--------------------------------------------------------------------------------------------------|
| Model Architecture       | 3D Convolutional Neural Network (3D-CNN)                                                         |
| Input Shape              | (20, 191, 121, 1)                                                                                |
| Additional Input Shape   | (20,65,121,1)                                                                                    |
| Output Shape             | (20, 191, 121, 1)                                                                                |
| Activation Functions     | LeakyReLU ( $\alpha = 0.01$ ), Tanh (output layer)                                               |
| Optimizer                | Adam ( $\alpha = 0.0001$ , $\beta_1 = 0.9$ , $\beta_2 = 0.999$ , $\epsilon = 1 \times 10^{-7}$ ) |
| Loss Function            | Mean Squared Error (MSE)                                                                         |
| Metrics                  | Accuracy                                                                                         |
| Batch Size               | 5                                                                                                |
| Epochs                   | 50                                                                                               |
| Number of GPUs           | 5 (using MirroredStrategy)                                                                       |
| Gradient Check Frequency | Every 5 epochs                                                                                   |

Author contributions. [Xiaohao Liu] was responsible for designing the research topic, collecting and conducting preliminary analysis of simulation data, leading the drafting of the manuscript, and overseeing subsequent revisions and improvements; [Zhaobin Li]assisted in

data validation and figure preparation, and provided key revision suggestions for the methodology section of the manuscript; [Xiaolei Yang]
took charge of the overall coordination of the research and funding support, reviewed the entire manuscript, mediated differences of opinion among authors, finalized the manuscript, and managed the submission process. All authors participated in discussions on key content of the manuscript and approved the final published version.

Competing interests. XY is a member of the editorial board of Wind Energy Science.

Acknowledgements. This work was supported by National Natural Science Foundation of China (NSFC) Excellence Research Group Pro-420 gram for "Multiscale Problems in Nonlinear Mechanics" (No. 12588201), the Strategic Priority Research Program, Chinese Academy of Sciences (CAS) (NO. XDB0620H0J), NSFC (NO. 12172360), and CAS Project for Young Scientists in Basic Research (YSBR-087).

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
