# Peer review of "PhyWakeNet: a dynamic wake model accounting for aerodynamic force oscillations"

_Wind Energy Science, 2025_

## Referee Comment (RC3)

**Review of the manuscript wes-2025-189, titled "PhyWakeNet: a dynamic wake model accounting for aerodynamic force oscillations" by X. Liu, Z. Li and X. Yang.**

This manuscript presents a physics-integrated generative machine learning model for predicting the wake velocity field of a wind turbine subjected to aerodynamic force oscillations. The model uses the mass and momentum conservation for the prediction of the time-average wake field, while a data-driven approach --using spatial and temporal modes of the flow field-- is used to model coherent velocity fluctuations that drive meandering. Small-scale turbulence in the flow field is generated with convolutional neural network that considers the mean flow field, wake meandering and inflow turbulence. Results show that the model can predict the temporal variations of the wake characteristics for the aerodynamic forcing with various frequencies with decent accuracy, demonstrating potential applications in wake management and mitigation.

The introduction and literature review are presented in the first part of the study, detailing prior efforts and the need for control of the wind turbine wakes. This is followed by a description of the methods used in the study, in which the time-averaged wake model, wake meandering models, and small-scale turbulence models are introduced along with their sub-models. Results from each of the models are then presented and compared with the data obtained from LES simulations. Moreover, the capability of the model to predict the instantaneous flow field and wake center is demonstrated. The manuscript is well structured and organized, and additional details are presented in the appendix. However, it may be helpful to add sentences in the main text that explicitly refer to the appendices. The figures are well prepared for clarity.

The manuscript presents a novel wake model capable of predicting the frequency-dependent variations in the wake characteristics. However, the study does not comment on the limitations of the data set and the presented model. Experimental studies from Messmer et al 2024 have shown that the frequency response of the wind turbine depends on the degree of freedom of the motion (e.g., side to side, front-back, etc.). While it can be inferred from the previous publications of the authors that the data used in this study and presented wake model is for side-to-side motion (or forcing), this has not been explicitly stated in the manuscript. Moreover, to facilitate the clarification of the derivation, a table or a figure showing the dependent and independent parameters for each of the wake models (or sub-models) would be helpful. In addition, there are several comments that need to be addressed.

1. Line 10: In the abstract, it is written that the result of turbulent kinetic energy is presented. However, no such results are presented in the manuscript. (only the variance of streamwise velocity fluctuations is presented). Please clarify or revise.
2. Check Line 32. Jensen is repeated with the reference.
3. Line 92: It is not clear how $\alpha$ has been defined. It would be nice to clarify this.

4. Line 132: Equations 13 and 3 appear to be defining the entrainment velocity. It would be nice if the differences between these equations are justified as one has the coefficient while the other does not.
5. Line 147: Lower and upper should be represented with the corresponding axes (y) as the upper and lower can also mean the boundaries in z-direction.
6. Line 160: Doesn't this suggest that the entrainment occurs mainly because of meandering or temporal variation of wake center location in time? How can this be justified in the near wake of a wind turbine, where meandering has not started yet?
7. Line 207: "Notably, the forcing term for the $i^{th}$ SPOD mode incorporates information from all considered SPOD modes' temporal derivatives $(\dot{a}_i)$ rather than relying solely on its own temporal derivative." It is not clear how this has been incorporated into the model. It would be nice to provide further details.
8. Figure 6: It would be nice to provide some details on how these values are evaluated using the model (listing all the inputs and the method used for the solution).
9. Figure 9: In the caption, "Figures d, e and i, j" is mentioned. However, the figures are missing from the figure. Moreover, the statement "The fourth and fifth rows correspond to two inflow conditions with $[I\infty = 0.8\%, L\infty = 1.0D]$ and $[I\infty = 0.2\%, L\infty = 4.0D]$, respectively." It is not related to the presented figures.
10. Line 275: Figure 11 is presented before figure 10. This can be revised.
11. Figure 11: x-label of the figures is not clear. It would be nice to clarify. Why are the normalized velocity deficit and the velocity variance being multiplied by constants?
12. Figure 12: Annotation and caption need to be corrected as there are missing sub-figures (d, e, i, j).
13. Figure 13: It would be nice to present the TI and integral length scales distinctly using larger spaces. Moreover, the turbulent length scales and integral length scales should be introduced before in the manuscript.
14. Line 335: Some comments about the generalizability of the model for other data sets (turbine models) or motion cases should be presented.

---

## Author Comment (AC1)

**Response to Reviewer 1**

Thank you for your time reviewing our manuscript. Your constructive comments have substantially improved the quality of our work. In the following, please find our point-to-point responses to your comments.

**General comments:** The authors present PhyWakeNet, a physics-integrated machine learning framework for dynamic wind turbine wake modeling under aerodynamic force oscillations. The model decomposes the instantaneous velocity field into time-averaged, meandering, and small-scale turbulent components. The time-averaged wake is governed by mass and momentum conservation with an entrainment-based closure; wake meandering uses conditional GAN-reconstructed SPOD modes with a data-driven dynamical system; small-scale turbulence is generated via a CNN. The model is trained and validated using LES data of a single NREL 5 MW turbine under transverse force oscillations at various Strouhal numbers ($St\_F$). It successfully captures frequency-dependent wake recovery, meandering amplitude, and turbulence statistics.

**Response:** Thank you for your kind consideration of our work.

**Comment 1**: Lack of Multi-Turbine Validation and Wake Superposition. Most analytical wake models fail precisely in wake interaction and superposition within wind farms — a critical practical challenge. Despite the stated motivation of "wake management in wind farms," all results are for a single turbine. No simulation or discussion addresses how PhyWakeNet handles partial wake overlap, merged wakes, or cumulative turbulence in arrays.

Why was no multi-turbine case investigated? This omission severely limits the claimed applicability. The authors should either: Include at least one 2–3 turbine inline or staggered case (with wake superposition), or explicitly justify the single-turbine focus and discuss planned extensions to farm-scale modeling. Without this, the wind farm relevance remains speculative.

**Response 1**: We fully agree that the development and validation of a wake superposition model for multi-turbine scenarios are crucial. To address this concern, in the revised manuscript (Appendix C), we added a test case with two turbines in an inline configuration, explained the model setup for this case, and discussed future development of the model for cases with more turbines and partial wakes. (lines 541-558, page 33)

**Comment 2:** Inadequate Literature Review on ML-Assisted Wake Modeling. The core innovation is ML integration (CGAN + CNN + physics) for dynamic wake prediction — yet the introduction lacks any review of prior ML-based wake or turbulence modeling. Relevant works are not cited. A dedicated paragraph is needed comparing: a) Data-driven vs. physics-constrained approaches; b) SPOD + GAN vs. POD-RBF, LSTM, or PINN methods; c) Quantitative performance (e.g., error in deficit, TKE, meandering). Without this context, the

novelty and improvement over existing ML wake models are unclear.

**Response 2:** Thanks for the suggestion. A dedicated paragraph has been added to the Introduction section to enhance the literature review (lines 53-72, page2-3). The novel contributions of the present work have been clarified (lines 73-80, page 3).

**Comment 3:** Figure Clarity and Completeness Issues. Figure 13: The difference between red and gray lines is not explained in the caption or text.

**Response 3:** The differences have been explained (lines 430-431, page 26).

**Comment 4:** Figure 13(c,d): Horizontal axis labels are illegible (overlapping or cut off).

**Response 4:** Corrected.

**Comment 5:** Figure 9 and Figure 12: Captions state "five rows (a–e, f–j)" but subplots d, e, i, j are missing in the figures.

**Response 5:** Corrected.

**Comment 6:** Figure 11: No legend — unclear which line corresponds to LES, PhyWakeNet, or submodels (only mentioned in the caption). These errors undermine result interpretation and must be corrected.

**Response 5:** Corrected.

**Comment 6:** Critical ML Methodology Relegated to Appendix. In data-driven modeling, dataset generation, model architecture, training strategy, and validation protocol are core contributions. Currently: a) LES setup, SPOD extraction, CGAN/CNN architectures, loss functions, training data split, and validation metrics are buried in appendices. b) The main text jumps from equations to results with minimal explanation of how the ML models were built or validated.

Move key ML details to the main body, including: a) Table of LES cases (St_F, turbulence intensity, length scale); b) CGAN and CNN architectures (layers, inputs, conditioning); c) Training/validation split, loss functions, and convergence; c) Number of SPOD modes (N) and sensitivity hyperparameter tuning may remain in appendix, but model design and data pipeline must be in the main paper.

**Response 6:** The suggested ML details have been moved to the main body, with the technical specifics (e.g., hyperparameter sensitivity) left in the appendices.

**Response 7:** Minor but Important Typos and Inconsistencies. Figure 1 caption: "GCAN" → should be CGAN.

**Response 7:** Corrected.

---

## Author Comment (AC2)

**Response to Reviewer 2**

Thank you for your time reviewing our manuscript. Your constructive comments have substantially improved the quality of our work. In the following, please find our point-to-point responses to your comments.

**General comments:** The manuscript presents a hybrid framework for developing a dynamic wake model that integrates first principles with machine learning (ML) techniques. The ultimate goal is to enable dynamic single-turbine wake modeling accounting for aerodynamic force oscillations. The authors leverage three sub-models to capture time-averaged wake, wake meandering, and small-scale wake turbulence. Some submodels are trained using LES data, and the results show generally good agreement with the ground truth. The work addresses an important problem in wind-energy modeling and has potential for impactful contributions. However, there are several points that need to be addressed to strengthen the manuscript.

**Response:** Thank you for your kind consideration of our work.

**Comment 1:** A primary concern is that the ML models are mostly treated as "black box". There is no discussion or demonstration of explainability, such as feature importance or SHAP tools. Since two of the submodels rely heavily on ML and directly affect the first submodel and overall wake predictions, the lack of explainability limits the credibility and understanding of the framework. It is strongly recommended to include an assessment of how the ML predictions are derived and how dependent they are on inputs.

**Response 1:** we have incorporated SHAP (SHapley Additive exPlanations) analysis to transition our CNN submodels from a "black box" to a more transparent system. Specifically, we added a new paragraph and a figure (Figure 10) that provides local SHAP explanations at two physically distinct locations: the wake centerline and the shear layer (lines 373-392, page 19-20). The SHAP analysis provides local interpretability for the CNN predictions, while global explainability remains a topic for future work.

**Comment 2:** The manuscript lacks a comprehensive review of recent literature in data-driven wake modeling, especially regarding generalization and explainability of ML-based wake predictions. Providing a critical comparison would clarify the novelty of this work.

**Response 2:** The literature review has been expanded (lines 53-72, page2-3). The novelty of this work has been clarified (lines 73-80, page 3).

**Comment 3:** Please provide a quantified comparison of the onset of wake meandering rather than relying on qualitative assessment from the results in Figure 12.

**Response 3:** A quantitative comparison has been provided (Figure 14; lines 421-424, page 24).

**Comment 4:** Regarding Figure 14: I agree with the authors that the model struggles more in

predicting the wake centerline velocity deficit than the wake center position. The model captures trends in wake centerline velocity deficit for $St_f = 0.12$ and $0.25$ but fails for $St_f = 0.84$ (in both wake centerline velocity deficit and position). The authors should discuss potential reasons for this discrepancy.

**Response 4:** Discussions on potential reasons have been added (lines 443-454 page 27).

**Comment 5:** The current model is developed and validated only for a single turbine. While the results are promising, it remains unclear how the approach would capture the cumulative effects of multiple interacting turbine wakes in a wind farm. This limitation should be explicitly discussed in the manuscript.

**Response 5:** The Reviewer raised an important concern. To address this concern, (1) we have added a test case with two turbines in an inline configuration (Appendix C), and (2) we have clarified the focus of the paper, which is on the dynamic wake model for a single wind turbine, and discussed the model development for wake overlap, merged wakes, or cumulative turbulence in arrays as future works (lines 540-558, page 33)

**Comment 6:** I believe there is a need to make lines 30–35 on page 2 more accurate.

**Response 6:** Specifics have been added in the revised text (lines 31-37, page 2 ).

**Comment 7:** Please clarify the definition of the turbine operational parameter Cop?

**Response 7:** Clarified. (lines 86-91, page 3).

**Comment 8:** Both the abstract and conclusion are highly qualitative. Including quantitative metrics on model accuracy would significantly improve the clarity and impact of the results.

**Response 8:** Included in both the abstract and conclusion (table 5, lines 8-11, page 1, lines 479-482, page 30).

**Comment 9:** Text in some sections seems repetitive, which disrupts readability. Streamlining text while keeping clarity is recommended.

**Response 9:** Streamlined with repetitive content removed.

**Comment 10:** There are a few citation issues (e.g., repeated names such as "Jensen Jensen") that should be corrected.

**Response 10:** Corrected.

**Comment 11:** Check Eq. (6): is it a complete equation or only a partial expression?

**Response 11:** Checked and revised to its complete form.(New Eq. (6))

**Comment 12:** Consider using a term other than "real" for LES predictions; "ground truth" is

more accurate.

**Response 12:** Revised to "ground truth".

**Comment 13:** Line 320, page 19, is unclear and needs clarification.

**Response 13:** Clarified (lines 458-465, page 27).

---

## Author Comment (AC3)

**Response to Reviewer 3**

Thank you for your time reviewing our manuscript. Your constructive comments have substantially improved the quality of our work. In the following, please find our point-to-point responses to your comments.

**General comments:** Review of the manuscript wes-2025-189, titled "PhyWakeNet: a dynamic wake model accounting for aerodynamic force oscillations" by X. Liu, Z. Li and X. Yang. This manuscript presents a physics-integrated generative machine learning model for predicting the wake velocity field of a wind turbine subjected to aerodynamic force oscillations. The model uses the mass and momentum conservation for the prediction of the time-average wake field, while a data-driven approach --using spatial and temporal modes of the flow field-- is used to model coherent velocity fluctuations that drive meandering. Small-scale turbulence in the flow field is generated with convolutional neural network that considers the mean flow field, wake meandering and inflow turbulence. Results show that the model can predict the temporal variations of the wake characteristics for the aerodynamic forcing with various frequencies with decent accuracy, demonstrating potential applications in wake management and mitigation. The introduction and literature review are presented in the first part of the study, detailing prior efforts and the need for control of the wind turbine wakes. This is followed by a description of the methods used in the study, in which the time-averaged wake model, wake meandering models, and small-scale turbulence models are introduced along with their sub-models. Results from each of the models are then presented and compared with the data obtained from LES simulations. Moreover, the capability of the model to predict the instantaneous flow field and wake center is demonstrated. The manuscript is well structured and organized, and additional details are presented in the appendix. However, it may be helpful to add sentences in the main text that explicitly refer to the appendices. The figures are well prepared for clarity. The manuscript presents a novel wake model capable of predicting the frequency-dependent variations in the wake characteristics.

**Response:** Thank you for your kind consideration of our work. The appendices have been referenced in the main text. (line 247, page 12; lines 291, page 14).

**Comment 1:** However, the study does not comment on the limitations of the data set and the presented model. Experimental studies from Messmer et al 2024 have shown that the frequency response of the wind turbine depends on the degree of freedom of the motion (e.g., side to side, front-back, etc.). While it can be inferred from the previous publications of the authors that the data used in this study and presented wake model is for side-to-side motion (or forcing), this has not been explicitly stated in the manuscript.

**Response 1:** The limitations of the data set and the presented model have been clarified in the revised manuscript (lines 487-490, page 30).

**Comment 2:** Moreover, to facilitate the clarification of the derivation, a table or a figure showing the dependent and independent parameters for each of the wake models (or sub-models) would be helpful. In addition, there are several comments that need to be addressed.

**Response 2:** A new Table (Table 1) has been added to show the dependent and independent parameters for the three sub-models.

**Comment 3:** Line 10: In the abstract, it is written that the result of turbulent kinetic energy is presented. However, no such results are presented in the manuscript. (only the variance of streamwise velocity fluctuations is presented). Please clarify or revise.

**Response 3:** It has been revised to variance of streamwise velocity fluctuations.

**Comment 4:** Check Line 32. Jensen is repeated with the reference.

**Response 4:** Corrected.

**Comment 5:** Line 92: It is not clear how $\alpha$ has been defined. It would be nice to clarify this.

**Response 5:** Clarified. (lines 122-123, page 5)

**Comment 6:** Equations 13 and 3 appear to be defining the entrainment velocity. It would be nice if the differences between these equations are justified as one has the coefficient while the other does not.

**Response 6:** Equation 13 denotes the instantaneous entrainment velocity across the wake boundary, which is computed from the velocity field predicted by the proposed model; while equation 3 relates the time-averaged entrainment velocity with the streamwise velocity difference between the ambient flow and the wake. (lines 326-335, page 16-17)

**Comment 7:** Line 147: Lower and upper should be represented with the corresponding axes (y) as the upper and lower can also mean the boundaries in z-direction.

**Response 7:** Revised as suggested. (lines 179, page 8)

**Comment 8:** Line 160: Doesn't this suggest that the entrainment occurs mainly because of meandering or temporal variation of wake center location in time? How can this be justified in the near wake of a wind turbine, where meandering has not started yet?

**Response 8:** Wake meandering increases the interface area between the wake and the ambient flow, and enhances the intensity of small scales around the wake-ambient flow interface. Thus, it is one key driver for the entrainment. In the near wake, the entrainment rate is approximated using that of the baseline case without aerodynamic force oscillations. The formulation has been revised to present the model properly. (lines 199-204, page 9).

**Comment 9:** Line 207: "Notably, the forcing term for the ith SPOD mode incorporates information from all considered SPOD modes' temporal derivatives ($\dot{a}$ ) rather than relying solely on its own temporal derivative." It is not clear how this has been incorporated into the model. It would be nice to provide further details.

**Response 9:** Further details have been provided. (lines 245-247, page 12)

**Comment 10:** Figure 6: It would be nice to provide some details on how these values are evaluated using the model (listing all the inputs and the method used for the solution).

**Response 10:** Details have been provided. (lines 338-342, page 17)

**Comment 11:** Figure 9: In the caption, "Figures d, e and i, j" is mentioned. However, the figures are missing from the figure. Moreover, the statement "The fourth and fifth rows correspond to two inflow conditions with [I∞ = 0.8%, L∞ = 1.0D] and [I∞ = 0.2%, L∞ = 4.0D], respectively." It is not related to the presented figures.

**Response 11:** Corrected.

**Comment 12:** Line 275: Figure 11 is presented before figure 10. This can be revised.

**Response 12:** Revised.

**Comment 13:** Figure 11: x-label of the figures is not clear. It would be nice to clarify. Why are the normalized velocity deficit and the velocity variance being multiplied by constants?

**Response 13:** Clarified. Multiplying constants is to ensure a clear visual comparison of quantities at different downstream locations within a single plot.

**Comment 14:** Figure 12: Annotation and caption need to be corrected as there are missing sub-figures (d, e, i, j).

**Response 14:** Corrected.

**Comment 15:** Figure 13: It would be nice to present the TI and integral length scales distinctly using larger spaces. Moreover, the turbulent length scales and integral length scales should be introduced before in the manuscript.

**Response 15:** Larger spaces have been used, and both have been introduced before figure 15 (the original figure 13) in the revised manuscript. (lines 292-297, page 14 )

**Comment 16:** Line 335: Some comments about the generalizability of the model for other data sets (turbine models) or motion cases should be presented. (lines xx-xx, page xx)

**Response 16:** Comments about the generalizability of the model have been added in the

Conclusion section of the revised manuscript. (lines 490-494, page 30)